# Rapid glycemic regulation in poorly controlled patients living with diabetes, a new associated factor in the pathophysiology of Charcot's acute neuroarthropathy

Dured Dardari[1,2,3]*, Georges Ha Van[4], Jocelyne M'Bemba[5], Francois-Xavier Laborne[6], Olivier Bourron[2,3,4], Jean Michel Davaine[2,7], Franck Phan[2,3,4], Fabienne Foufelle[2], Frédéric Jaisser[2], Alfred Penfornis[1,8], Agnes Hartemann[2,3,4]

1 Department of Diabetes, Sud Francilien Hospital Center, Corbeil-Essonnes, France, 2 INSERM UMRS 1138, Cordeliers Research Center, Paris, France, 3 Sorbonne University, Paris, France, 4 Department of Diabetes, Pitié-Salpêtrière Hospital, Paris, France, 5 Podology Unit, Cochin Hospital, Paris, France, 6 Clinical Research Unit, Sud Francilien Hospital Center, Corbeil-Essonnes, France, 7 Department of Vascular Surgery Pitié-Salpêtrière Hospital, Paris, France, 8 Paris-Sud Medical School, Paris-Saclay University, Corbeil-Essonnes, France

* dured.dardari@gmail.com

**Data Availability Statement:** All relevant data are within the paper.

## Abstract

### Objective

Aggressive antidiabetic therapy and rapid glycemic control are associated with diabetic neuropathy. Here we investigated if this is also the case for Charcot neuroarthropathy.

### Research design and methods

HbA1c levels and other relevant data were extracted from medical databases of 44 cases of acute Charcot neuroarthropathy.

### Results

HbA1c levels significantly declined from 8.25% (67mmol/mol) [7.1%–9.4%](54-79mmol/mol), at -6 months (M-6), to 7.40%(54mmol/mol) [6.70%–8.03%] (50–64 mmol/mol) during the six months preceding the diagnosis of Charcot neuroarthropathy (P <0.001).

### Conclusions

HbA1c levels significantly declined during the six months preceding the onset of Charcot neuroarthropathy. This decline seems to be a associated factor with the appearance of an active phase of Charcot neuroarthropathy in poorly controlled patients with diabetic sensitive neuropathy.

**Funding:** The authors received no specific funding for this work.

**Competing interests:** The authors have declared that no competing interests exist.

**Abbreviations:** CEREES, Committee of Expertise for Research Studies and Evaluations in the Field of Health; CN, Charcot neuroarthropathy; INDS, National Institute of Health Data; OAD, oral antidiabetic; OPG, osteoprotegerin; RANKL, receptor activator of nuclear factor-B; T1DM, type 1 diabetes mellitus; T2DM, type 2 diabetes mellitus; TIND, treatment-induced neuropathy of diabetes.

## Introduction

Aggressive antidiabetic therapy and rapid glycemic control may result in a diabetic neuropathy called treatment-induced neuropathy of diabetes (TIND) [1, 2]. On the other hand, Charcot neuroarthropathy is a chronic, devastating and destructive disease of the bone structure and joints in patients with neuropathy. It is characterised by painful or painless bone and joint destruction in limbs that have lost sensory innervation [3, 4]; we have reported a case developed following rapid glycemic control [5]. In this project, we accessed databases of three diabetes centres, extracted a clinically significant number of cases of Charcot neuroarthropathy, and investigated whether its onset was associated with rapid glycemic control. We aimed to investigate this association for better understanding of the physiology of this complication.

## Research design and methods

We retrospectively analyzed medical records of persons with diabetes and acute Charcot neuroarthropathy, who had been referred to three hospital centres (Department of Diabetes, Sud-Francilien Hospital, Corbeil-Essonnes, France; Podology Unit, Cochin Hospital, Paris, France; and Department of Diabetes, Pitie-Salpetriere Hospital, Paris) from 2008 to 2018. Patients with suspected active Charcot neuroarthropathy were referred by general practitioners, diabetologists or emergency services of these centres.

### Patients

The inclusion criteria were: (i) a diagnosis of acute Charcot neuroarthropathy based on the clinical presentation of a hot and red foot with associated joint oedema and confirmed by radiological examination and nuclear magnetic résonance (MRI), and (ii) patients with HbA1c values (measured in their usual laboratories or at the hospital) at 6 and 3 months before the diagnosis of Charcot neuroarthropathy, and at the time of diagnosis.

The type of Charcot neuroarthropathy was determined according to the classification of Sanders and Frykberg [6, 7]. In Practice, The data of the radiology departments from already mentioned hospitals were screened in order to identify patients having practiced MRI with the diagnosis of active phase neuroarthropathy, the authors then included only patients with sensitive neuropathy and having performed three HbA1c tests within the time specified in the inclusion criteria, the clinical records were consulted for this information. (S1 Fig).

Out of 59 files screened by the three radiology departments, 44 files compliant with the inclusion criteria were finally included (the necessary population size according to the workforce calculation used in the statistical method was 43 patients). It is also important to note that patients with a diagnosis of OAN in active phase can also perform their MRI outside the hospital radiology department), all data were monitored by the clinical research unit of the Centre hospitalier sud Francilien, all three radiology departments used the classification of Eichenholtz [8] to determine an active phase of OAN, (MRI showing signs of microfracture, oedema of the bone marrow and bone contusion to determine an active phase of OAN.

### Study design

The study adhered to the Declaration of Helsinki and the state laws of France. According to the national law and the research guidelines of local hospitals, a retrospective study with treatment administered as a part of routine clinical practice does not require the approval of an institutional review board. According to the French law (Decree No. 2016–1872 of December 26, 2016), a file was submitted to the INDS (Institut National des Données de Santé; National Institute of Health Data) and the CEREES (Comité d'Expertise pour les Recherches, les Etudes

et les Evaluations dans le domaine de la Santé; Committee of Expertise for Research Studies and Evaluations in the Field of Health).

For each record selected, one of us (DD) collected data in an Excel file using a password-protected computer, without Internet access.

### Statistical analysis

To demonstrate an HbA1c reduction of at least 2 points (within six months before the onset of Charcot neuroarthropathy) with an estimated variance of 2.5, we hypothesized that a population size of 43 patients was needed. Statistical analyses were performed with R software (version 3.6.1, The R Foundation for Statistical Computing, Vienna, Austria).

Data are presented as median and interquartile ranges. Statistical analysis for numeric variables was performed according to the non-parametric Friedman and Wilcoxon tests. Pairwise comparisons between time levels were performed using the Bonferonni-Holms method for p-value adjustment. Categorical variables were compared using Fisher's exact test. Kernel density estimates were used to investigate multimodality [9].

## Results

A total of 44 patients diagnosed with diabetes and acute Charcot neuroarthropathy were included in the study. Patients median age was 60 years (interquartile range = 50–71 years); 29 were males and 15 females living with diabetes for 16±3,2 years. Among patients with a foot deformity, 29 had a distortion in the tarsometatarsal joints and 13 in the metatarsophalangeal joints; the remaining two had knee or ankle disease. All subjects had diabetic sensitive neuropathy; only three patients had reported a previous autonomic neuropathy diagnosed before the onset of acute Charcot neuroarthropathy.

### HbA1c levels

HbA1c levels declined from 8.25% (67mmol/mol) [7.1%–9.4%](54-79mmol/mol), at -6 months (M-6), to 7.80%(62mmol/mol) [7.15%–8.25%](55-67mmol/mol) at -3 months (M-3) and 7.40%(54mmol/mol) [6.70%–8.03%] (50–64 mmol/mol) at the time of diagnosis of Charcot neuroarthropathy (M0) (S1 Fig). This decrease in HbA1c level was highly significant (p <0.001).

Of the 44 included patients, 15 presented with T1DM and 29 with TD2M (S1 Table). HbA1c levels significantly decreased in patients with T1DM (-1.78% p <0.01) but not in patients with T2DM (-0.74%, not significant).(S2 Fig).

### Recent intensification of treatment

Thirty-three patients underwent an intensification of antidiabetic treatment (16 with insulin, fou with liraglutide and six with oral antidiabetics (OADs). Intensification of antidiabetic treatment was unknown for seven patients) and one patient received a pancreas transplant (S1 Table). Of the 16 patients receiving insulin intensification, ten had T1DM, and six had T2DM (S1 Table).

Significant heterogeneity was found in the magnitude of HbA1c reduction with treatment intensification. Kernel density estimates [9] showed that the HbA1c reduction data (x-axis intervals of 0.2%) was well fitted with two normal distributions, one for the mean HbA1c reduction = 0.86% ± 1.66% and the second one for the HbA1c reduction = 4.23% ± 3.61%.

## Discussion

In our study, HbA1c levels significantly declined during the six months before the onset of acute Charcot neuroarthropathy. HbA1c levels decreased more in patients with T1DM compared to patients with T2DM (S2 Fig). Significant heterogeneity was found in the magnitude of HbA1c reduction with treatment intensification (S3 Fig). Taking a cut-off of 2 points for the decrease in HbA1c% in six months, seventeen participants (two with T1DM and fifteen with T2DM) had an HbA1c reduction equal to or higher than the cut-off.

We included 44 diabetic patients with acute Charcot neuroarthropathy, a large sample size for an unusual disorder (its prevalence in patients with diabetes is 0.1% and 0.4% [10,11]). As expected, all included subjects had diabetic sensitive neuropathy. The distribution of HbA1c reduction among the included patients was bimodal. Therefore, we used a cut-off point (HbA1c reduction of more than 2 points in six months) similar to the one used by Gibbons et al. [1, 2] to define TIND (HbA1c reduction of more than 2 points in three months) [1]. In our study, we have a similar reduction, but in over six months, the clinical presentation of CN is longer than that of TNID.

Interestingly, the prevalence of TIND reported by Gibbons et al. (up to 10%) [1, 2] is similar to ours (11.4% of treatment-associated acute Charcot neuroarthropathy).

The Charcot neuroarthropathy (CN) is a rare and devastating disorder presenting with peripheral and/or autonomic neuropathy that affects people with diabetes [3, 4], [in our study only three patients had reported a diagnosis of autonomic neuropathy]. A late diagnosis of CN can have serious consequences, including amputation. The physiopathology of this disorder is poorly known. A new series of experiments was carried on the evolution of bone modelling factors in the appearance of Charcot neuroarthropathy with the implication of the receptor activator of nuclear factor-B ligand (RANKL) and its natural antagonist, osteoprotegerin (OPG) [12,13], it has already been described the value of OPG can vary after insulin treatment in patients living with type 1 diabetes [14], this may refine the hypotheses and the results of our study, however, the RANKL value was not evaluated in that study [14].

More, a diagnosis of neuroarthropathy in particular circumstances which can be linked to rapid correction of the glycemic balance without treating the subject directly has been described, as the onset of Charcot neuroarthropathy during weight loss after bariatric surgery [15] and after kidney and pancreas transplantation [16], it is for the first time we can clearly describe an association type relationship and not a direct causality between the onset of acute CN with the rapid glycemic control, a study tackled the level of HbA1C as a predictive factor in the onset of an OAN, but without conclusive results [17], in other words, one thinks that this rapid reduction in HbA1C levels contributes to the month in a partial way to the activation of the inflammatory phenomena which are currently the triggers of OAN [18]

Study limitations include: (i) The single-group design often employed in retrospective studies limits the researchers' ability to determine cause and effect. Although it is usually impossible to include a control group in a retrospective study, whenever possible a control group should be included to help establish the cause-and-effect relationship; (ii) data extraction was centre-based, potentially ignoring site-specific factors.

In conclusion, HbA1c levels significantly declined during the six months preceding the onset of acute Charcot neuroarthropathy. This prevalence is similar to the prevalence of up to 10% reported by Gibbons et al. [1, 2] for TIND. Our results suggest that rapid glycemic control in chronically unbalanced patients with diabetic neuropathy is an associated factor for acute Charcot neuroarthropathy.

## Supporting information

**S1 Fig. Decrease in HbA1c levels before the onset of Charcot neuroarthropathy.** Values are given as median [interquartile ranges]. Statistical significance was tested with the non-parametric Friedman's test.
(DOCX)

**S2 Fig. Reduction in HbA1c levels according to the type of diabetes.**
(DOCX)

**S3 Fig. Heterogeneity of the magnitude of HbA1c reduction in the 6 months preceding the onset of Charcot neuroarthropathy.** (non parametric Friedman test).
(DOCX)

**S1 Table. Therapeutic aspects of the included patients.**
(DOCX)

**S1 Image. Fow chart of collecting data.**
(DOCX)

## Acknowledgments

The authors acknowledge Ricardo P. Garay (Craven, Villemoisson-sur-Orge, France) for editorial support.

## Author Contributions

**Conceptualization:** Dured Dardari, Frédéric Jaisser, Alfred Penfornis, Agnes Hartemann.

**Data curation:** Jocelyne M'Bemba, Francois-Xavier Laborne.

**Investigation:** Georges Ha Van, Olivier Bourron, Jean Michel Davaine.

**Methodology:** Franck Phan.

**Project administration:** Fabienne Foufelle.

**Supervision:** Dured Dardari.

**Validation:** Dured Dardari.

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
