## [Decision Letter · Decision Letter 0]

1 Apr 2020

PONE-D-20-06583

Rapid glycemic regulation in poorly controlled patients living with diabetes, a new approach <gwmw class="ginger-module-highlighter-mistake-type-3" id="gwmw-15856881875558206946191">in</gwmw> understanding the pathophysiology of Charcot's acute <gwmw class="ginger-module-highlighter-mistake-type-1" id="gwmw-15856881875558947104717">neuroarthropathy</gwmw>

PLOS ONE

Dear dr dardari,

Thank you for submitting your manuscript to PLOS ONE. After careful consideration, we feel that it has merit but does not fully meet PLOS ONE’s publication criteria as it currently stands. Therefore, we invite you to submit a revised version of the manuscript that addresses the points raised during the review process.

We would appreciate receiving your revised manuscript by May 16 2020 11:59PM. To enhance the reproducibility of your results, we recommend that if <gwmw class="ginger-module-highlighter-mistake-type-6" id="gwmw-15856881964705953311038">applicable you</gwmw> deposit your laboratory protocols in protocols.io, where a protocol can be assigned its own identifier (DOI) such that it can be cited independently in the future. For <gwmw class="ginger-module-highlighter-mistake-type-6" id="gwmw-15856881969980246782953">instructions see</gwmw>: http://journals.plos.org/plosone/s/submission-guidelines#loc-laboratory-protocols

A rebuttal letter that responds to each point raised by the academic editor and reviewer(s). This letter should be uploaded as <gwmw class="ginger-module-highlighter-mistake-type-3" id="gwmw-15856881991579041326485">separate file</gwmw> and labeled 'Response to Reviewers'.A marked-up copy of your manuscript that highlights changes made to the original version. This file should be uploaded as <gwmw class="ginger-module-highlighter-mistake-type-3" id="gwmw-15856882007947672661342">separate file</gwmw> and labeled 'Revised Manuscript with Track Changes'.An unmarked version of your revised paper without <gwmw class="ginger-module-highlighter-mistake-type-3" id="gwmw-15856882013257735270713">tracked</gwmw> changes. This file should be uploaded as <gwmw class="ginger-module-highlighter-mistake-type-3" id="gwmw-15856882018130528188147">separate file</gwmw> and labeled 'Manuscript'.

Please <gwmw class="ginger-module-highlighter-mistake-type-6" id="gwmw-15856882027153945484978">note while</gwmw> forming your response, if your article is accepted, you may have the opportunity to make the peer review history publicly available. The record will include editor decision letters (with reviews) and your responses to reviewer comments. If eligible, we will contact you to opt in or out.

We look forward to receiving your revised manuscript.

Kind regards,

Manal S. <gwmw class="ginger-module-highlighter-mistake-type-1" id="gwmw-15856882053484411390607">Fawzy</gwmw>, Ph.D., M.D.

Academic Editor

PLOS ONE

Journal Requirements:

1. In ethics statement in the manuscript and in the online submission form, please provide additional information about the patient records used in your retrospective study. Specifically, please ensure that you have discussed whether all data were fully anonymized before you accessed them and/or whether the IRB or ethics committee waived the requirement for informed consent. If patients provided informed written consent to have data from their medical records used in research, please include this information.

2. Thank you for including your funding statement; "not applicable"

Additional Editor Comments (if provided):

Based on the constructive and valued reviewers’ comments, the authors should address, in addition, the following concerns:

- Provide more information about the inclusion and exclusion process, screening of cases etc. A flow chart would be most helpful as the patient-group in question is a heterogenous one and too little data are presented regarding this issue.

- Provide a fully plausible and validated biological model for the described association. Otherwise, the authors should point to this issue in the study shortcomings for the readers.

- Acknowledge the fact that data speak describe an association and not a causality.

- Sufficiently address in their study limitations all the possible pitfalls that stem from drawing conclusions from retrospective data.

Reviewers' comments:

Reviewer's Responses to Questions

**Comments to the Author**

1. Is the manuscript technically sound, and do the data support the conclusions?

Reviewer #1: Partly

Reviewer #2: Yes

2. Has the statistical analysis been performed appropriately and rigorously? 

Reviewer #1: I Don't Know

Reviewer #2: Yes

3. Have the authors made all data underlying the findings in their manuscript fully available?

The PLOS Data policy requires authors to make all data underlying the findings described in their manuscript fully available without restriction, with rare exception (please refer to the Data Availability Statement in the manuscript PDF file). The data should be provided as part of the manuscript or its supporting information, or deposited <gwmw class="ginger-module-highlighter-mistake-type-3" id="gwmw-15856882206899974495972">to</gwmw> a public repository. For example, in addition to summary statistics, the data points behind means, medians and variance measures should be available. If there are restrictions on publicly sharing data—e<gwmw class="ginger-module-highlighter-mistake-type-6" id="gwmw-15856882223699625479975">.</gwmw>g. <gwmw class="ginger-module-highlighter-mistake-type-1" id="gwmw-15856882229505033897090">participant</gwmw> privacy or use of data from a third party—those must be specified.

Reviewer #1: Yes

Reviewer #2: Yes

4. Is the manuscript presented in an intelligible fashion and written in standard English?

PLOS ONE does not copyedit accepted manuscripts, so the language in <gwmw class="ginger-module-highlighter-mistake-type-3" id="gwmw-15856882249455246828644">submitted</gwmw> articles must be clear, correct, and unambiguous. Any typographical or grammatical errors should be corrected at revision, so please note any specific errors here.

Reviewer #1: Yes

Reviewer #2: Yes

5. Review Comments to the Author

Please use the space provided to explain your answers to the questions above. You may also include additional comments for the author, including concerns about dual publication, research ethics, or publication ethics. <gwmw class="ginger-module-highlighter-mistake-type-6" id="gwmw-15856882280618318727862">(</gwmw>Please upload your review as an attachment if it exceeds 20,000 characters)

Reviewer #1: To the authors:

Thank you for the opportunity to review this <gwmw class="ginger-module-highlighter-mistake-type-1" id="gwmw-15856882294543587272824">intersting</gwmw> paper by Dardari et al.

The paper touches upon a very interesting and largely overlooked problem in <gwmw class="ginger-module-highlighter-mistake-type-1" id="gwmw-15856882307373516530623">glycaemic</gwmw> regulation, namely the complications that can arise from too rapid/aggressive blood glucose treatment.

While the issues regarding cardiovascular and retinal disease in this regard <gwmw class="ginger-module-highlighter-mistake-type-3" id="gwmw-15856882316763224509431">is</gwmw> <gwmw class="ginger-module-highlighter-mistake-type-1" id="gwmw-15856882316764421058823">wellknown</gwmw>, the problems with <gwmw class="ginger-module-highlighter-mistake-type-1" id="gwmw-15856882316763873544039">neuropathic</gwmw> patients are more elusive.

Below, I have listed my comments on the paper as a whole. Unfortunately, my copy does not contain line <gwmw class="ginger-module-highlighter-mistake-type-6" id="gwmw-15856882331351905120624">numbering and</gwmw> so I cannot address my comments to any specific line.

1) More information about the data collection and quality of the database would be helpful. Considering the time period and multiple sites, one wonders about the <gwmw class="ginger-module-highlighter-mistake-type-1" id="gwmw-15856882346266953796452">homogenicity</gwmw> of the charts/electronic systems. How were data logged and retrieved?

2) In the same <gwmw class="ginger-module-highlighter-mistake-type-6" id="gwmw-15856882366210285016844">vein it</gwmw> would be helpful if the paper included a <gwmw class="ginger-module-highlighter-mistake-type-1" id="gwmw-15856882366218889101846">flowchart</gwmw> of patients screened, excluded and so on. How were the data <gwmw class="ginger-module-highlighter-mistake-type-1" id="gwmw-15856882384425905003395">entried</gwmw> validated/controlled? How was the diagnosis confirmed?

The 44 cases have been gathered over a 10 year time period, which makes me think that a lot of borderline cases were removed (unless we're talking about smaller hospitals). Thus, it appears the authors had a good and solid screening of cases. However, this is not apparent in the manuscript.

3) What timeline was used to give the diagnosis acute Charcot foot? When was it considered chronic instead? What was the temperature spread? Any bilateral feet? Recurrences?

4) What was the spread in dates from HbA1c analyses at each time point?

5) Since all the patients had Charcot, they probably all had neuropathy before onset. Is it possible that only 3 cases with registered painful neuropathy is due to insufficient data quality.

6) Were there any of the cases with other <gwmw class="ginger-module-highlighter-mistake-type-6" id="gwmw-15856882457538292147058">possible plausible</gwmw> listed causes of Charcot foot? For instance trauma, physical stress or surgery?

7) It would be helpful to list HbA1c values in IFCC units as well as DCCT.

8) Did every single patient decrease in HbA1c, or did someone increase as well? If so, were there any differences between the two groups?

9) A change from 66.7 mmol/<gwmw class="ginger-module-highlighter-mistake-type-1" id="gwmw-15856882488674427369829">mol</gwmw> to 57.4 mmol/<gwmw class="ginger-module-highlighter-mistake-type-1" id="gwmw-15856882488672406722665">mol</gwmw> is relatively small, and unavoidable in clinical practice. Do you have any indication about <gwmw class="ginger-module-highlighter-mistake-type-3" id="gwmw-15856882503155184797133">frequency</gwmw> of Charcot arthropathy being associated with such a change in HbA1c? Do you have any suggestions about how to effectively mitigate this issue in high risk patients?

10) Why did the patients get a better HbA1c before Charcot onset? You write that 26<gwmw class="ginger-module-highlighter-mistake-type-6" id="gwmw-15856882522164014636389">(</gwmw>27) of 44 patients with altered medication – what happened to the rest? Maybe expand the table to say a bit more about these cases. Did they for instance get hospital control instead of GP control? Did they change lifestyle/lose weight? Is it possible to elaborate further since it's very important?

11) It seems that <gwmw class="ginger-module-highlighter-mistake-type-3" id="gwmw-15856882552605569563017">data</gwmw> support an association to better regulation especially in T1DM group. What happens if you remove the T2DM patients from the analysis completely. Why do you think this is the case? Could it be a different <gwmw class="ginger-module-highlighter-mistake-type-3" id="gwmw-15856882564132405105517">mechanisms</gwmw>?

12) The authors have found an interesting association between acute Charcot foot and <gwmw class="ginger-module-highlighter-mistake-type-2" id="gwmw-15856882572213929706917">resent</gwmw> decrease in HbA1c. But of course association does not mean causality. Could you speculate about other factors that might contribute to this <gwmw class="ginger-module-highlighter-mistake-type-2" id="gwmw-15856882582869979896254">find</gwmw>?

Would you be able to design a model to predict or adjust for any such confounding factors to your dataset?

13) At the bottom of page 8, the authors write that the reduction is mainly due to insulin intensification. Has any analysis done <gwmw class="ginger-module-highlighter-mistake-type-3" id="gwmw-15856882602814557116714">of</gwmw> this and/or i.e. <gwmw class="ginger-module-highlighter-mistake-type-1" id="gwmw-15856882605807033556421">contributing</gwmw> factors?

14) Do you know if any of the patients complained about TIND?

Reviewer #2: Introduction

1- Definition of Charcot neuroarthropathy is needed

2- Clear statement to describe why this research is important.

Methods

1-More justification why the authors select <gwmw class="ginger-module-highlighter-mistake-type-3" id="gwmw-15856882635276213516370">retrospective approach</gwmw> and more information about selection of participants

Findings and discussion

Can the authors discuss more about the contributions the study makes to existing knowledge or literature?

6. PLOS authors have the option to publish the peer review history of their article (what does this mean?). If published, this will include your full peer review and any attached files.

If you choose “no”, your identity will remain <gwmw class="ginger-module-highlighter-mistake-type-6" id="gwmw-15856882670023865267001">anonymous but</gwmw> your review may still be made public.

Reviewer #1: No

Reviewer #2: No

<gdiv></gdiv>

---

## [Author Response · Author response to Decision Letter 0]

16 Apr 2020

Dear Editor, Dear Reviewer

First of all, I want to thank you for your support for the potential publication of our manuscript. We have taken into account all of your relevant remarks, with particular emphasis on the fact that we are describing an association and not a causality. Presenting our responses and a new version of our manuscript, I will be happy to consider your future comments.

Kindly regards 

Dured Dardari

Additional Editor Comments 

Based on the constructive and valued reviewers’ comments, the authors should address, in addition, the following concerns:

1- Provide more information about the inclusion and exclusion process, screening of cases etc. A flow chart would be most helpful as the patient-group in question is a heterogeneous one, and too little data are presented regarding this issue.

Answer from Dardari

I hope the modifications made in the Research Design and Methods respond to your important remarks.

2- Provide a fully plausible and validated biological model for the described association. Otherwise, the authors should point to this issue in the study shortcomings for the readers. Acknowledge the fact that data speak describe an association and not a causality.

Answer from Dardari

We have emphasized that our results represent an association and not a direct causality in the discussion section.

3- Sufficiently address in their study limitations all the possible pitfalls that stem from drawing conclusions from retrospective data.

Answer from Dardari

Done.

Reviewer #1

1) More information about the data collection and quality of the database would be helpful. Considering the time period and multiple sites, one wonders about the homogeneity of the charts/electronic systems. How were data logged and retrieved?

2) In the same vein, it would be helpful if the paper included a flowchart of patients screened, excluded, and so on. How were the data entered validated/controlled? How was the diagnosis confirmed?

The 44 cases have been gathered over a 10 year time period, which makes me think that a lot of borderline cases were removed (unless we are talking about smaller hospitals). Thus, it appears that the authors had a good and solid screening of cases. However, this is not apparent in the manuscript

Answer from Dardari

The data from the radiology departments of 3 hospitals were questioned in order to identify the patients who performed MRI during the period selected with the diagnosis of active phase neuroarthropathy. The authors then took into account only the patients having a sensitive neuropathy according to the information in the medical file and a presence of the three assays of HbA1c performed in the times indicated in the inclusion criteria. In this sense, 59 files were delivered by the three radiology departments, 44 files corresponding to the inclusion criteria were finally included (the number necessary according to the workforce calculation used in the statistical method for 43 patients). It is also important to note that patients with a diagnosis of OAN in active phase prove also perform their MRI outside the hospital radiology department, all data were monitored by r the clinical research unit of the southern francilein hospilatier centre.

3) What timeline was used to give the diagnosis of acute Charcot foot? When was it considered chronic instead? What was the temperature spread? Any bilateral feet? Recurrences?

Answer from Dardari

All three radiology departments used the classification of Eichenholtz (8) to determine an active phase of OAN. MRI was used to determine an active phase of OAN: signs of microfracture, oedema of the bone marrow, and bone contusion. 

4)What was the spread in dates from HbA1c analyses at each time point?

Answer from Dardari

HbA1c levels declined from 8.25% [7.1%–9.4%] at -6 months (M-6), to 7.80% [7.15%–8.25%] at -3 months (M-3) and 7.40% [6.70%–8.03%] at the time of diagnosis of Charcot neuroarthropathy (M0) (Supplementary Figure 2). 

5) Since all the patients had Charcot, they probably all had neuropathy before onset. Is it possible that only 3 cases with registered painful neuropathy are due to insufficient data quality?

Answer from Dardari 

We wanted to know if the patients described the presence of painful neuropathy in the moment of the presentation of acute Charcot. Only three patients had already known neuropathy-related pain. Unfortunately, we do not have more information on this issue.

6) Were there any of the cases with other possible plausible listed causes of Charcot's foot? For instance, trauma, physical stress or surgery?

Answer from Dardari 

As you have pointed out, it is believed that the rapid reduction of an HbA1c is associated with the other factors already described in the physiopathology of the OAN so we studied only the patient files looking for the main criterion of our study.

7) It would be helpful to list HbA1c values in IFCC units as well as DCCT

Answer from Dardari 

Thank you for this comment, we have given the values in IFCC.

8) Did every single patient decrease in HbA1c, or did someone increases as well? If so, were there any differences between the two groups?

Answer from Dardari 

Significant heterogeneity was found in the magnitude of HbA1c reduction with treatment intensification (Figure 1). Kernel density estimates showed that the HbA1c reduction data (x-axis intervals of 0.2%) was well fitted with two normal distributions, one of mean HbA1c reduction = 0.86% ± 1.66% and the second one of HbA1c reduction = 4.23% ± 3.61%. Taking a cut-off of 2 points for the decrease in HbA1c% in six months, twenty participants had an HbA1c reduction equal to or higher than the cut-off (Figure 3). 

9) Change from 66.7 mmol/mol to 57.4 mmol/mol is relatively small, and unavoidable in clinical practice. Do you have any indication about frequency of Charcot arthropathy being associated with such a change in HbA1c? Do you have any suggestions about how to effectively mitigate this issue in high risk patients?

Answer from Dardari 

After the publication of this retrospective study, we wish to set up a prospective evaluation of the bone modulating factors under similar conditions, which one can say that the return towards a “normal level of HbA1c In patients with a chronic glycemic balance must be progressive in the presence of peripheral neuropathy (+/-osteoporosis as a high risk example), this have to be indicated even if the patients have no retinal complication or other reason to indicate a gradual return, it may also be thought that preventive measures prove to be recommended to patients considered at risk, such as adapting footwear, etc., there is a real need to educate clinicians on this therapeutic aspect.

10) Why did the patients get a better HbA1c before Charcot onset? You write that 26(27) of 44 patients with altered medication – what happened to the rest? Maybe expand the table to say a bit more about these cases. Did they for instance get hospital control instead of GP control? Did they change lifestyle/lose weight? Is it possible to elaborate further since it is very important?

Answer from Dardari 

The clear and precise information on the therapeutic evolution was obtained only for the patients noted in the table. We do not have information on this question.

11) It seems that data support an association with better regulation, especially in the T1DM group. What happens if you remove the T2DM patients from the analysis completely. Why do you think this is the case? Could it be different mechanisms?

Answer from Dardari 

We added the differential measure between type 1 and type 2, the potency on HbA1c is probably linked to the effectiveness of insulin therapy in reducing the level of HbA1c, however on think that the impact of the correction of HbA1c on the causation of the OAN is the same in the two types of diabetes.

12) The authors have found an interesting association between acute Charcot foot and a recent decrease in HbA1c. Nevertheless, of course, association does not mean causality. Could you speculate about other factors that might contribute to this find?

Would you be able to design a model to predict or adjust for any such confounding factors to your dataset?

 Answer from Dardari

W.J. Jeffcoate Theories concerning the pathogenesis of the acute Charcot foot suggest that trauma is the basis for triggering inflammatory factors in OAN. We share this opinion, although not all patients with diabetes and trauma develop OAN, so the association of our results and already known factors appears to be interesting and justified.

13) At the bottom of page 8, the authors write that the reduction is mainly due to insulin intensification. Has any analysis done of this and/or, i.e. contributing factors?

Answer from Dardari

We wrote this sentence because the reduction in HbA1c levels was greater in patients with T1D.

14) Do you know if any of the patients complained about TIND? 

Answer from Dardari

Unfortunately, we do not have this information.

Reviewer #2

1) Definition of Charcot neuroarthropathy is needed

Answer from Dardari

 Many thanks for your suggestion; the modification was done.

2) A clear statement to describe why this research is important

Answer from Dardari

Done. 

3) More justification why the authors select retrospective approach and more information about selection of participants

Answer from Dardari

Taking into account the reduced prevalence of this devastating complication, we opted for a retrospective study first in order to prove the concept. We will be delighted to inform you of the results of our prospective phase.

4) Can the authors discuss more the contributions the study makes to existing knowledge or literature?

Answer from Dardari

Done.

---

## [Decision Letter · Decision Letter 1]

30 Apr 2020

Rapid glycemic regulation in poorly controlled patients living with diabetes, a new associated factor in the pathophysiology of Charcot's acute neuroarthropathy

PONE-D-20-06583R1

Dear Dr. dardari,

We are pleased to inform you that your manuscript has been judged scientifically suitable for publication and will be formally accepted for publication once it complies with all outstanding technical requirements.

With kind regards,

Manal S. Fawzy, Ph.D., M.D.

Academic Editor

PLOS ONE

Additional Editor Comments (optional):

The authors have adequately addressed the concerns raised by the reviewers.

Reviewers' comments:

Reviewer's Responses to Questions

**Comments to the Author**

1. If the authors have adequately addressed your comments raised in a previous round of review and you feel that this manuscript is now acceptable for publication, you may indicate that here to bypass the “Comments to the Author” section, enter your conflict of interest statement in the “Confidential to Editor” section, and submit your "Accept" recommendation.

Reviewer #1: All comments have been addressed

Reviewer #2: All comments have been addressed

2. Is the manuscript technically sound, and do the data support the conclusions?

Reviewer #1: Yes

Reviewer #2: Yes

3. Has the statistical analysis been performed appropriately and rigorously? 

Reviewer #1: Yes

Reviewer #2: Yes

4. Have the authors made all data underlying the findings in their manuscript fully available?

Reviewer #1: Yes

Reviewer #2: Yes

5. Is the manuscript presented in an intelligible fashion and written in standard English?

Reviewer #1: Yes

Reviewer #2: Yes

6. Review Comments to the Author

Reviewer #1: (No Response)

Reviewer #2: The authors have addressed my previous comments and I am happy with their responses. I accept this paper to be published.

7. PLOS authors have the option to publish the peer review history of their article (what does this mean?). If published, this will include your full peer review and any attached files.

Reviewer #1: No

Reviewer #2: Yes: Amer Al-Sahouri

<gdiv></gdiv>

---

## [Editor Report · Acceptance letter]

7 May 2020

PONE-D-20-06583R1 

Rapid glycemic regulation in poorly controlled patients living with diabetes, a new associated factor in the pathophysiology of Charcot's acute neuroarthropathy 

Dear Dr. dardari:

I am pleased to inform you that your manuscript has been deemed suitable for publication in PLOS ONE. Congratulations! Your manuscript is now with our production department. 

With kind regards,

on behalf of

Professor Manal S. Fawzy 

Academic Editor

PLOS ONE